# Perceptions Related to Nursing and Nursing Staff in Long-Term Care Settings during the COVID-19 Pandemic Era: Using Social Networking Service

**DOI:** 10.3390/ijerph18147398

**Published:** 2021-07-11

**Authors:** Juhhyun Shin, Sunok Jung, Hyeonyoung Park, Yaena Lee, Yukyeong Son

**Affiliations:** 1College of Nursing, Ewha Womans University, Seoul 03760, Korea; okijung1@naver.com (S.J.); hy_geul@naver.com (H.P.); yaena1224@gmail.com (Y.L.); 2Samsung Seoul Hospital, Seoul 06351, Korea; elly2kevin@naver.com

**Keywords:** nursing staff, COVID-19, nursing homes, Social Network Service, big-data

## Abstract

Purpose: The purpose of this study was to investigate what opinions and perceptions people have about nursing and the role of nursing staff in nursing homes (NHs) on Social Networking Service (SNS) by analyzing large-scale data through social big-data analysis. Methods: This study investigated changes in perception related to nursing and nursing staff in NHs during the COVID-19 pandemic era using target channels (blogs, cafes, Instagram, communities, Twitter, etc.). Data were collected on the channel from 12 September 2019 to 11 September 2020, 6 months before and after 12 March 2020 when the COVID-19 pandemic was declared. Selected keywords included “nursing,” “nurse,” and “nursing staff,” and included words were “long-term care settings,” “geriatric hospital,” and “nursing home.” Text mining, opinion mining, and social network analysis were conducted. Results: After the COVID-19 pandemic, the frequency of keywords increased about 1.5 times compared to before. In March 2020 when the COVID-19 pandemic was declared, the negative phrase “be infected” ranked number one, resulting in a sharp 8% rise in the percentage of negative words in that month. The related words that have risen in rank significantly, or were newly ranked in the Top 30 after the pandemic, were related with COVID-19. Conclusion: The public began to realize the role of nursing staff in the prevention and management of mass infection in NHs and the importance of nursing staff after the pandemic. Further studies should examine the perceptions of those who have received nursing services and include a wide range of foreign channels.

## 1. Introduction

### 1.1. Background

Coronavirus disease 19 (COVID-19) began in Wuhan, China in December 2019, and 167 million confirmed cases were reported in about 221 countries, of which 3.47 million people died, and the number of confirmed cases worldwide increased sharply by May 2021 [1]. Accordingly, the World Health Organization (WHO) declared COVID-19 a pandemic on 11 March 2020 due to its widespread capability and high fatality rate [2]. In Korea, after the first confirmed case on 20 January 2020, a massive community infection occurred in religious facilities and geriatric hospitals, starting with 31 confirmed cases on 18 February 2020, and as of May 2021, the total number of confirmed cases reached 136,467, with 1934 deaths [3]. Especially, late adults and elders (people aged 50–64) increased their mortality rate by 30 times when they were infected with COVID-19, 90 times when they were 65–74 years old, 220 times when they were 75–84 years old, and 630 times when they were 85 years old or older [4]. In Korea, by May 2021, the proportion of those aged 80 or older among COVID-19 confirmed cases was 4.15%, while the proportion of those aged 80 or older among COVID-19 deaths was 55.33% [3]. In addition, the older the age, the higher the fatality rate [3].

The proportion of the elderly population (65 years or older) in Korea was 16.5% in 2021, and Korea is expected to enter a super-aged society by 2050 [5,6]. The number of nursing homes (NHs) has increased significantly since the implementation of long-term care insurance for the elderly in 2008 [7]. The number of long-term care institutions providing facility benefits increased by 54% from 3751 in 2010 to 5806 in 2021 [8], and there are 1461 geriatric hospitals [9]. Concerns are growing over the increase in the number of COVID-19 cases in NHs, one of the most vulnerable facilities to the COVID-19 crisis. There were 132,305 COVID-19 deaths and 652,476 COVID-19-confirmed cases in U.S. NHs as of 9 May 2021 [10]. The proportion of deaths related to long-term care facilities by country was about 36% in the United States, 75% in Australia, 64% in New Zealand, and 59% in Spain, showing that more than 50% of COVID-19 deaths were residents of long-term care facilities in many countries [11]. The number of COVID-19 deaths in Korean NHs and geriatric hospitals was 563 out of 1486 total COVID-19 deaths, accounting for 37.8% [12].

During the COVID-19 outbreak, nursing staff, the only licensed healthcare workforce among NH staff, have played an important role in providing professional nursing care and skills for residents’ physical and psychological needs, as well as meeting needs for high-risk elderly people with chronic diseases [13]. Recently, in the United States, research reported that more nursing staff was related with lower fatality rates of COVID-19-confirmed residents, and the role and importance of nursing staff in NHs was emphasized [14,15,16,17], suggesting that more registered nurses (RNs) had less infection and mortality rates with COVID-19. Subsequently, public opinion that requires appropriate resources and placement of nursing staff in nursing home after COVID-19 is increasing. Additionally, in Korea, studies on the importance of nursing staff and the need to increase nurse staffing levels in NHs have continued [18,19], but it has not been practically introduced due to poor working conditions and support. In the current situation, where the importance of nursing staff and nursing is being mentioned due to the COVID-19 crisis, this study investigated what opinions and perceptions people have about nursing and the role of nursing staff in Korean NHs.

Recently, studies to predict trends using social big data are being conducted as unstructured data in the form of text produced in online channels has a very high impact on the actual economy and society [20]. The traditional research methods using surveys obtain information through limited questions and sampling, so they have difficulties in that the reliability and validity of the analyzed data must be verified from various perspectives. On the other hand, social big data provides a much larger amount of data from various people [21], making it more accurate to identify the public perception. Therefore, it is easy to identify consumers’ needs in terms of healthcare service through social big data [22]. In the quality of care, it is important to identify the perceptions and needs of consumers to reflect their needs in nursing services [23]. Research has been conducted on laws and policy-improvement measures for the use of big data [24], analysis of the specialized rehabilitation treatment health insurance fee [25], and death rates for elderly people with dementia [26], all using the big data model in the healthcare field in the post-coronavirus era [27] in Korea. However, big-data analysis research on NHs is very limited not only in Korea, but also in other countries. In addition, as a result of a literature review on a paper applying social network analysis (SNA) among nursing research literature published from 1965 to 2017 by searching databases (Ovid Healthstar, CINAHL, PubMed Central, Scopus) and hard copy literature, the use of SNA in nursing research is rare, although it first appeared in 1995. This is because it is unknown to most nursing scholars [28]. Therefore, this study provides base data for evaluating the usefulness of big-data utilization in nursing through social big-data analysis, which has not been significantly used in nursing research. We tried to investigate the spread of creativity and knowledge of nursing practices through understanding the roles and importance of nursing practices by identifying the perception of nursing and nursing staff in NHs before and after the COVID-19 pandemic using target channels in Korea (blogs, cafes, Instagram, communities, Twitter, etc.).

### 1.2. Research Purpose

The purpose of this study was to investigate public opinions and perceptions about nursing and the role of nursing staff in NHs on SNS by analyzing large-scale data through social big-data analysis.

## 2. Methods

### 2.1. Design

A cross-sectional design was used.

### 2.2. Sampling and Data Collection

The data were collected from 12 September 2019 to 11 September 2020, 6 months before and after 12 March 2020 when the COVID-19 pandemic was declared. The vaccine was not initiated in Korea during the data-collection period. Data collection was conducted from target channels (blogs, cafes, Instagram, communities, Twitter, etc.) using a collection engine (robot), such as crawlers, without considering the days of the week, weekends, and holidays. Target channels were about 7000 online channels including major communities (e.g., Bobaedream, Boombu, and DC Inside), blogs, and cafes (e.g., Naver, Daum). Retweet posts on Twitter were excluded from the collection because they could overlap and affect the results. This study utilized people’s general perceptions through keyword search in English and Korean via online channels. The selected keywords were “nursing” OR “nurse” OR “nursing staff,” and the included words were “long-term care settings” OR “geriatric hospital” OR “nursing home.” By collecting keywords and included words together, only posts in which the included words and keywords appeared at the same time were collected and analyzed. Through this method, it is possible to narrow the scope of the analysis target to nursing and nursing staff in long-term care facilities, by entering keywords at the same time.

### 2.3. Data Analysis

#### 2.3.1. Data Cleansing

Social big-data text analysis was conducted with Sometrend. Sometrend was used as analysis algorithms (dig index) based Hadoop big-data framework, and equipped with robots and discourse-analysis (keyword and sentiment analysis) technology. Duplicate and similar documents were removed to increase the accuracy of the collected data. As a result, “be dazed,” which greatly affected the analysis results, was designated as an exclusion word due to overlapping posts made by the same person. In addition, advertising documents were also removed, especially for blog data.

#### 2.3.2. Text Mining

Text mining refers to extracting useful information from unstructured text written in human language using natural language-processing (NLP) technology. In other words, text mining refers to discovering the hidden meaningful information of big data, such as classifying, clustering, or summarizing by grasping the linkage of unstructured texts [29]. Text mining was conducted on the collected online documents related to perception of nursing staff and nursing in NHs during the COVID-19 crisis. The collected documents were processed in natural language through morpheme analysis, part of speech tagging and post processing, syntax chunking, syntax analysis, and semantic analysis. Afterwards, entity names, nouns, and predicates were extracted using the entity name, dictionary, and thesaurus.

The frequency of keywords was analyzed via text mining. Frequency of keywords by channel and weekly frequency changes were analyzed through the number of posts including the keywords in the collection channels.

#### 2.3.3. Opinion Mining

Opinion mining analyzes users’ opinions (positive, normal, negative, etc.) by applying NLP technology and emotional-analysis technology to text sentences on social media [29], and is also called buzz analysis in marketing [30]. In this study, the classification of emotional words was completed using the sentiment dictionary, developed by the Vaiv company to which Sometrend belongs. The dictionary was developed through a process of morpheme analysis, part of speech tagging, syntax analysis, and semantic analysis by domain. It considered the meaning of terms by using large-scale semantic classification system. Emotional analysis related to nursing staff was conducted to identify emotional words that are highly related to the keywords and analyze differences before and after the COVID-19 pandemic, monthly differences in the frequency ranking of emotional words, and the ratio of each polarity (positive, negative, neutral).

#### 2.3.4. SNA

SNA aims to analyze the network connection structure and connection strength to determine which messages are propagated through which route and to whom they can affect [29]. This study detected issues through information extraction based on SNA that analyzed future trends such as flows and patterns. Semantic network analysis helps researchers discover the structure of text by measuring the co-occurrence of specific words. Co-occurrence increases as the connection between words strengthens, and the value of co-occurrence ranges from 0 to 12 [31].

Therefore, related word analysis based on social data was conducted to identify the top 25 words with high frequency appearing with keywords before and after the COVID-19 pandemic, respectively. It was conducted through the process of extracting related words having a meaningful relationship with keywords by syntax analysis based on core NLP. In addition, the top 10 words by month were identified to discover the change of related words over time. We visualized the results through Tag Cloud and Word Cloud to clearly identify and effectively communicate differences in the appearance frequency of the related words.

### 2.4. Ethical Consideration

The IRB was exempt because this study did not have participants but used open-access data (Approval Number. 136-4).

## 3. Results

### 3.1. Textmining

#### 3.1.1. Frequency of Keyword by Channel

Table 1 shows the frequency of keyword-related searches by channel before and after the COVID-19 pandemic. The table shows that the total number (7343) of keyword-related searches increased about 1.5 times after the COVID-19 pandemic compared to before (4959). The increase on Twitter increased significantly when looking at the increase by channel, and the search frequency ranking was the same before and after the pandemic in the order of blog, Instagram, community, and Twitter.

#### 3.1.2. Changes in Frequency of Keyword by Week

Figure 1 shows the number of keyword-related searches by channel by week. The total number of searches increased sharply between the second week of February (162 cases) and the fourth week of February (334 cases), and the second week of March (400 cases) and the third week of March (831 cases). By channel, community showed the highest in the third week of March (121 cases), Instagram in the second week of June (95 cases), blog in the third week of March (597 cases), and Twitter in the third week of March (67 cases). In other words, the number of keyword-related searches has increased overall since 1 month before the COVID-19 pandemic declaration, and it increased sharply shortly after the second week of March when the COVID-19 pandemic was declared. The total number of searches decreased sharply from more than 800 to 353 2 weeks after the pandemic declaration (fourth week of March), has continued to decline slightly, and recently has remained at 200 to 300 without showing significant changes over time.

### 3.2. Opinion Mining

#### 3.2.1. Frequency Ranking of Keyword-Related Emotional Words and Comparison of Emotional Word Rates Before and After the COVID-19 Pandemic

Table 2 and Table 3 show the frequency ranking of keyword-related emotional words before and after the COVID-19 pandemic. Emotional words that ranked in the top 10 both before and after the pandemic were ‘necessary,’ ‘diverse,’ ‘good,’ ‘possible,’ ‘assistance,’ ‘assist,’ ‘require,’ and ‘hard,’ showing no considerable difference before and after the pandemic. Most emotional words in the top 10 were positive or neutral words, which seems to be a result of the promotional or informational documents about NHs with contents such as ‘recommendation of good facilities,’ ‘assist of nursing staff,’ and ‘comfortable facilities’ rather than direct perceptions of nursing and nursing staff in NHs.

Looking at the pre-pandemic period results, it seems the negative words ‘inconvenient’ and ‘difficult’ also resulted from informational and promotional documents about NHs with contents such as ‘facilities for the elderly who are difficult or inconvenient to move by oneself.’ Other negative words mentioned in relation to NH nursing staff were used in context such as ‘the difficulties of working in NHs,’ ‘the difficulties of supplying medical masks due to the spread of COVID-19,’ ‘the pain of a nursing care worker facing the death of elderly in NH,’ ‘worries about adapting to work while looking for a geriatric hospital,’ and ‘violence between patients in a mental geriatric hospital leading to emergence of problem of shortage of nursing staff.’

According to the post-pandemic period results, the percentage of positive words decreased by 4%, as Figure 2 shows, and the negative word ‘infected’ newly appeared in the top 10 and topped the list. The positive word ‘fast,’ which appeared in the top 10, would not have significant meaning, as promotional or informational documents made up the majority. Other negative words mentioned in relation to NH nursing staff were used in context such as ‘the difficulties of working in NHs,’ ‘concern and anxiety about mass infection in NHs,’ and ‘criticism of care worker who refused government requests and attended church.’

#### 3.2.2. Emotional Word Ranking Related to Keyword and Comparison of Emotional Word Rates by Month

Table 4 shows the frequency ranking of keyword-related emotional words by month. When the COVID-19 pandemic was declared in March 2020, the negative word ‘infected,’ which had not appeared in the top 10 before, ranked number one, and it seems to have resulted in a sharp 13% drop in the proportion of positive words and a sharp 8% increase in the proportion of negative words in that month (see Figure 3).

### 3.3. SNA

#### 3.3.1. Frequency Ranking of Keyword-Related Word Before and After the COVID-19 Pandemic

Table 5 presents a list of keyword-related words in order of frequency. Related words that commonly showed a high frequency before and after the COVID-19 pandemic were ‘recuperation,’ ‘hospital,’ ‘geriatric hospital,’ ‘NH,’ ‘facility,’ ‘society,’ ‘treatment,’ ‘dementia,’ and ‘hospitalization.’ These words are related to the general attributes of NHs and can be seen as not having significant meaning. Related words that have dropped significantly after the pandemic include words such as ‘support,’ ‘program,’ ‘home-living,’ ‘prevention,’ ‘visiting nursing,’ and ‘visiting care.’ Related words that have risen significantly or newly ranked in the top 25 after the pandemic were ‘corona,’ ‘infection,’ ‘corona 19,’ ‘mass infection,’ ‘virus,’ ‘area,’ ‘outcome,’ ‘coronavirus,’ ‘infectious disease,’ and ‘guardian.’ Figure 4 and Figure 5 presents visualized results of keyword-related words and differences in the appearance frequency of them before and after the COVID-19 pandemic.

#### 3.3.2. Frequency Ranking of Keyword-Related Words by Month

As shown in Table 6, keyword-related words that have a connection with COVID-19 appeared in the top 10 around March 2020, when the COVID-19 pandemic was declared. However, after 2 months, the ranking did not differ much from before the pandemic.

## 4. Discussion

This study compared public perceptions of NH nursing staff before and after the COVID-19 pandemic (before any vaccines were initiated) using text and opinion mining and SNA. First, as a result of text mining, the total keyword-related searches increased by 50% after the COVID-19 pandemic. Especially after looking at the changes in search frequency by week, the number of keyword-related searches has increased overall since 1 month before the COVID-19 pandemic declaration, and increased sharply shortly after the second week of March when the COVID-19 pandemic was declared. It could mean that, as the number of COVID-19 patients in NHs sharply increased after the first confirmed case of COVID-19 in NHs on March 4, 2020 [19], the interest in nursing staff who prevent and manage infections in NHs increased. We found that the frequency of searches related to NH nursing staff was closely related to the trend of COVID-19. The rapidly increasing keyword frequency decreased 2 weeks after the COVID-19 pandemic declaration, maintaining a constant level of frequency without significant changes. The public, unlike experts, perceive new, difficult to control, severe consequences, and larger exposure sizes as more dangerous [32]. It can be interpreted that the public’s risk awareness was very high in March 2020 due to the unprecedented situation of the COVID-19 pandemic. In addition, as COVID-19 entered a phase of sedation, it affected the psychological aspects of the public that COVID-19 could be overcome over time, which could be interpreted as a decrease in the number of keywords due to lower risk recognition than the point of the pandemic. As such, data generated online, including SNS, is useful in identifying social issues and public interests, and related research is actively being conducted [33,34,35]. The media’s impact on how the public perceive nursing and nursing staff in modern society is significant [36], and images of nursing staff are formed not only through direct contact with nursing staff, but also through the media. Therefore, it is necessary to continuously monitor the public’s perception of nursing staff on SNS.

Second, as a result of opinion mining, there were many positive or neutral words overall, and words such as ‘necessary,’ ‘good,’ and ‘assistance’ were common before and after the COVID-19 pandemic. The frequency rankings of emotional words before and after the pandemic were similar. The positive and neutral words likely derived from promotional and informative documents on NHs before and after the pandemic, which seemed to be less related to COVID-19. However, the phrase ‘be infected,’ which had not previously appeared, topped the list in March 2020, at the time of the COVID-19 pandemic, with the percentage of negative words rising sharply (by 8%), and documents expressing concern and anxiety about mass infection in NHs also increased. This result may have occurred due to a sharp increase in the number of documents, including the phrase ‘be infected,’ due to the first major outbreak of mass infection in NHs. Since then, the proportion of negative words seemed to remain similar to before the pandemic as the contents of negative words before the pandemic were replaced with those related to infection. The infection-related phrase emerged during this pandemic era, which the SNS rarely expressed. This result suggests there is a lot of news related to COVID-19 in the mass media. In addition, before the pandemic, general difficulties with NH staff and in the supply and demand of goods in the early stages of COVID-19 accounted for a large portion of negative words, but after the pandemic, documents related to COVID-19 infection in NHs and difficulties with NH staff increased significantly. Nursing staff who directly face and take care of the elderly are forced to play sacrificial roles and keep responsibilities in order to effectively cope with the problems caused by the new respiratory infectious disease [37]. Therefore, further research on the roles of nursing staff handling infection control in long-term care settings should be conducted by analyzing large-scale data posted in the form of social media channels.

The related words, which newly appeared after the COVID-19 pandemic, were ‘corona,’ ‘infection,’ ‘mass infection,’ ‘virus,’ and ‘infectious disease,’ which showed high frequencies when the pandemic was declared in March 2020, but 2 months later showed similar frequencies as before the pandemic. In other words, the words related to COVID-19 showed a higher frequency than words related to NH after the pandemic. This result means that the public’s interest shifted from the environment and service of NHs and routine nursing services to concerns about infection in NHs and the role of nursing staff in the COVID-19 pandemic era, as they realized the vulnerability of infection in NHs and importance of nursing staff.

Since the outbreak of COVID-19, a number of group infections have been reported in NHs, accounting for 12.9% of all group infections, and the fatality rate of related people is 12.0% [38]. The government raised the COVID-19 crisis level to serious and focused on pan-governmental disinfection. In the case of NHs and geriatric hospitals, a large number of deaths occurred due to the sealed space and characteristics of the vulnerable elderly; therefore, the government tried to reduce the occurrence of confirmed cases by blocking the loop of group infection in the facilities [38]. This situation emphasized the importance of RNs’ roles in preventing and managing infections, as studies showed that the proportion of nursing staff had to do with infection control interest, willingness to improve infection control, practice of infection control, and monitoring activities, and that NHs with higher nurse staffing levels had a lower number of COVID-19-confirmed cases [39,40]. In addition, nursing staff’s awareness and knowledge of infection control can prevent the greater spread of infection, as nursing staff with many opportunities to contact patients or residents directly or indirectly can be infected and spread infections to residents [19].

Nonetheless, the ongoing spread of COVID-19 in NHs shows the urgent need for infection control in NHs where elderly people reside in groups. In other words, this pandemic has highlighted the difficulties, institutional limitations, and problems within the healthcare field that have emerged since COVID-19, and the importance of nursing staff at the center of COVID-19 has increased more than ever. The findings in this study support this public perception. Furthermore, it is necessary to adapt to the current situation and introduce new measures. Since February 2020, the government has prohibited visits to NH facilities to inhibit unnecessary human contact [41]. The restrictions on visiting and group activities can negatively affect patients’ and residents’ mental and physical health [42]. It is time to consider the long-term care setting in the pandemic era, which permits family members to meet in a safe environment and provides a professional visiting guide, infection-control system for visitors, and end-of-life care [43].

Healthcare industries are producing and managing huge amounts of big data to meet present and future social needs [44]. Social big-data analysis can be used to understand the general public’s perceptions or social problems to propose policies by checking the thoughts and responses of individuals rapidly circulating online in real time [45]. Therefore, stepping stones should be created through the use and analysis of big data produced through SNS to improve problems and prepare policies related to NHs, which should be able to grasp the public’s perception. In this study, for example, we identified increased public concern about mass infection in NHs and nursing staff’s difficulties in coping with the COVID-19 crisis. Through the information, we are able to propose policies to improve NH nursing staff’s situations, such as systemic infection-prevention education and legal staffing levels in NHs. In addition, this study offers base data necessary for improving the NH system to respond to the spread of new infectious diseases in the future by getting an insight into pending issues.

This study searched nursing and nursing staff at NHs using big-data analysis sites. Some keywords were related to promotional or informational phrases, even though duplicate, similar, and spam documents were removed and refined results were analyzed. The original documents were checked prior to interpretation of the collected data, but there is a limit to generalizing the study’s results. Nevertheless, this study is significant in that it grasped public perception using big-data analysis that is still in its infancy in the nursing area.

This study has some limitations, as the data collection was completed before COVID-19 vaccines were initiated. In Korea, preemptive tests have been conducted on residents and workers in NHs since December 21, 2020, and those groups are now vaccinated [46]. The proportion of confirmed COVID-19 cases in NHs and geriatric hospitals among total confirmed cases was 5.6% right after vaccination, but recently that number has decreased significantly to 2% [47]. However, some problems have arisen due to the continued emergence of post-vaccination deaths, including cases in NHs [48]. More timely research using SNS is required in the current situation. Although this study collected data from various channels, there may be data bias in that the data were analyzed based on selected data. Unlike quantitative research, there is a limitation in that different results can be derived depending on the data-processing source and sentiment dictionary due to the characteristics of unstructured data.

## 5. Conclusions

This study compared public perceptions of nursing staff in NHs before and after the COVID-19 pandemic using text and opinion mining and SNS. Unlike before the COVID-19 pandemic, people began to realize the role of RNs in the prevention and management of mass infection in NHs, and the importance of nursing staff increased after the pandemic. This study investigated the perception of nursing staff in NHs through SNS data of unspecified individuals in Korea. Further research should be conducted to investigate global perceptions of long-term care settings during and after the COVID-19 pandemic era.

## Figures and Tables

**Figure 1 ijerph-18-07398-f001:**
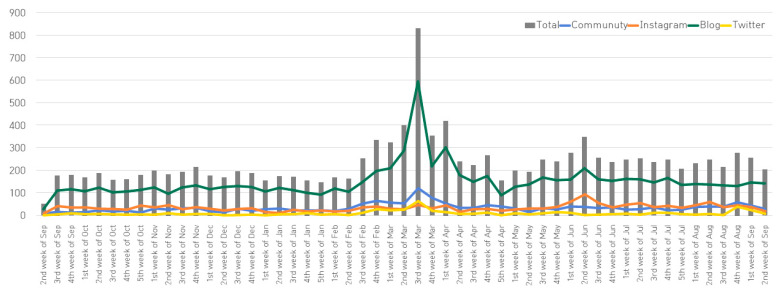
Frequency by channel and week related to keyword.

**Figure 2 ijerph-18-07398-f002:**
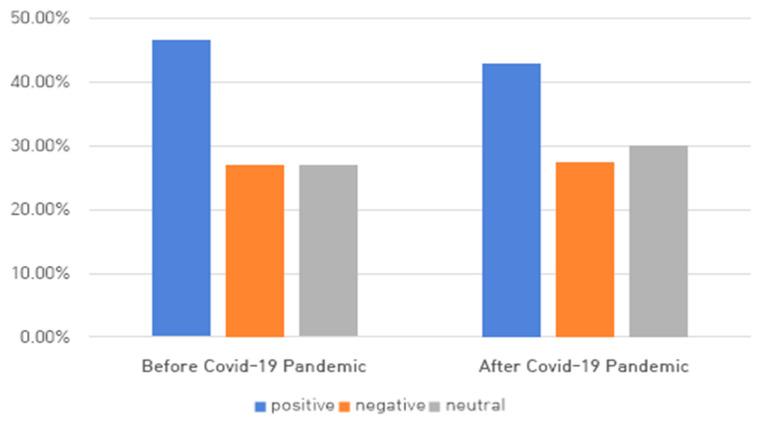
Emotional word rates before and after the COVID-19 pandemic.

**Figure 3 ijerph-18-07398-f003:**
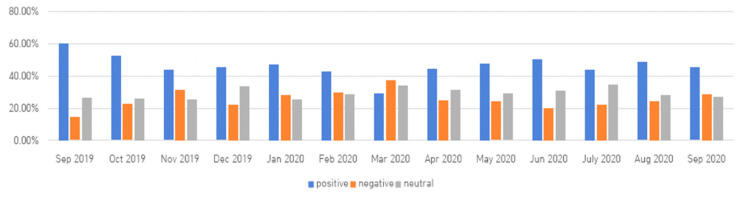
Emotional word rates by month.

**Figure 4 ijerph-18-07398-f004:**
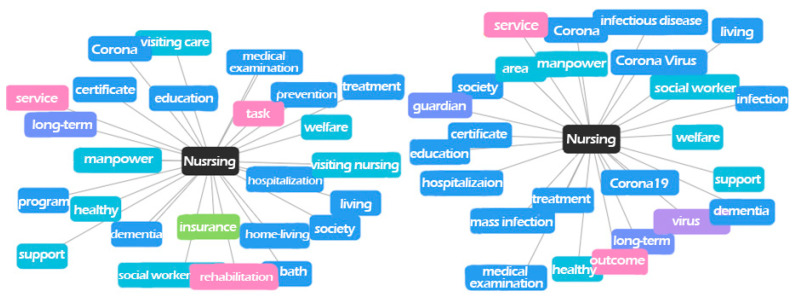
Keyword-related words before (**left**) and after (**right**) the COVID-19 pandemic.

**Figure 5 ijerph-18-07398-f005:**
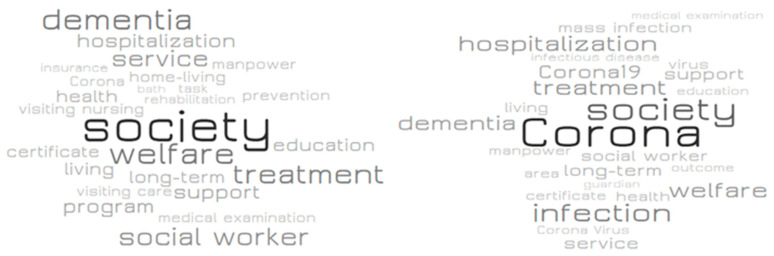
Word cloud for keyword-related words before (**left**) and after (**right**) the COVID-19 pandemic.

**Table 1 ijerph-18-07398-t001:** Frequency of keyword by channel before and after COVID-19 pandemic.

	Community		Instagram		Blog		Twitter		Total	
	Before Pandemic	After Pandemic	Before Pandemic	After Pandemic	Before Pandemic	After Pandemic	Before Pandemic	After Pandemic	Before Pandemic	After Pandemic
Total	706	1082	783	1098	3258	4797	212	366	4959	7343

**Table 2 ijerph-18-07398-t002:** Top 10 keyword-related emotional words in frequency ranking before and after the COVID-19 pandemic.

Rank	Before COVID-19 Pandemic			After COVID-19 Pandemic		
1	necessary	252	N	be infected	346	−
2	diverse	245	N	necessary	341	N
3	good	225	+	diverse	324	N
4	be possible to	170	+	good	248	+
5	assistance	160	+	assistance	235	+
6	assist	106	N	fast	216	+
7	require	99	N	be possible to	210	+
8	inconvenient	94	−	assist	189	N
9	the best	89	+	require	156	N
10	be hard	82	−	be hard	138	−

Note. N: neutral, +: positive, −: negative.

**Table 3 ijerph-18-07398-t003:** Top 10 keyword-related emotional words in frequency ranking by each emotion before and after the COVID-19 pandemic.

Rank	Before COVID-19 Pandemic			Rank	After COVID-19 Pandemic		
	+	−	N		+	−	N
1	good	inconvenient	necessary	1	good	be infected	necessary
2	be possible to	be hard	diverse	2	assistance	be hard	diverse
3	assistance	difficult	assist	3	fast	inconvenient	assist
4	the best	lack	require	4	be possible to	difficult	require
5	convenient	pain	isolation	5	hope	concern	be high
6	fast	concern	be high	6	the best	pain	isolation
7	be good	insufficient	new	7	be good	be difficult to	differ
8	happy	be difficult to	differ	8	convenient	anxiety	understand
9	safety	violence	be large	9	healthy	side effect	new
10	relief	be inconvenient	important	10	safety	worry	exact

Note. N: neutral, +: positive, −: negative.

**Table 4 ijerph-18-07398-t004:** Top 10 keyword-related emotional words in frequency ranking by month.

**Rank**	**September 2019**		**October 2019**		**November 2019**		**December 2019**		**January 2020**		**February 2020**		**March 2020**	
1	diverse	N	diverse	N	diverse	N	necessary	N	necessary	N	necessary	N	be infected	−
2	be possible to	+	good	+	good	+	diverse	N	diverse	N	good	+	isolation	N
3	fast	+	necessary	N	necessary	N	be high	N	good	+	be possible to	+	necessary	N
4	good	+	assistance	+	be possible to	+	be possible to	+	assistance	+	diverse	N	assistance	+
5	necessary	N	be possible to	+	assist	N	good	+	be possible to	+	isolation	N	be exposed	N
6	assist	N	require	N	inconvenient	−	be few	N	assist	N	assistance	+	exact	N
7	clean	+	assist	N	require	N	require	N	isolation	N	lack	-	fast	+
8	assistance	+	inconvenient	-	be good	+	new	N	be assaulted	−	require	N	good	+
9	be healthy	+	be hard	-	assistance	+	be good	+	be hard	−	the best	+	the best	+
10	safety	+	the best	+	be hard	−	inconvenient	−	difficult	−	be good	+	contact with	N
**Rank**	**April 2020**		**May 2020**		**June 2020**		**July 2020**		**August 2020**		**September 2020**			
1	fast	+	diverse	N	diverse	N	necessary	N	hope	+	necessary		N	
2	necessary	N	necessary	N	good	+	diverse	N	diverse	N	good		+	
3	diverse	N	good	+	necessary	N	be possible to	+	necessary	N	assist		N	
4	be infected	−	assistance	+	be possible to	+	assist	N	be possible to	+	diverse		N	
5	good	+	be possible to	+	assistance	+	be high	N	good	+	differ		N	
6	assistance	+	assist	N	require	N	good	+	fast	+	inconvenient		-	
7	be hard	−	fast	+	inconvenient	−	require	N	assistance	+	be good		+	
8	be possible to	+	understand	N	assist	N	assistance	+	assist	N	be possible to		+	
9	remember	N	require	N	be high	N	inconvenient	−	require	N	be hard		−	
10	contact	N	inconvenient	−	fast	+	understand	N	be hard	−	require		N	

Note. N: neutral, +: positive, −: negative.

**Table 5 ijerph-18-07398-t005:** Keyword-related word ranking before and after the COVID-19 pandemic.

Rank	Before Covid-19 Pandemic		After Covid-19 Pandemic	
	Related Word	Frequency	Related Word	Frequency
1	society	986	Corona	1525
2	welfare	673	society	1201
3	dementia	664	infection	971
4	treatment	629	treatment	834
5	social worker	564	hospitalization	833
6	service	529	welfare	805
7	support	462	dementia	789
8	program	460	Corona19	718
9	hospitalization	451	support	622
10	education	428	long-term	620
11	long-term	414	service	615
12	living	413	social worker	584
13	health	408	mass infection	559
14	certificate	391	health	548
15	home-living	365	living	545
16	manpower	353	virus	528
17	prevention	351	area	502
18	visiting nursing	347	manpower	500
19	visiting care	337	certificate	495
20	Corona	334	outcome	474
21	medical examination	325	Corona Virus	474
22	task	316	education	469
23	rehabilitation	312	infectious disease	444
24	insurance	307	medical examination	434
25	bath	284	guardian	424

Note. Dark gray: A rise in rank, Light gray: A fall in rank.

**Table 6 ijerph-18-07398-t006:** Keyword-related word ranking by month.

**Rank**	**Spetember 2019**	**October 2019**	**November 2019**	**December 2019**	**January 2020**	**February 2020**	**March 2020**
1	society	society	society	society	society	society	Corona
2	dementia	dementia	dementia	dementia	welfare	Corona	infection
3	service	social worker	welfare	welfare	service	welfare	Corona19
4	social worker	welfare	social worker	service	dementia	treatment	mass infection
5	treatment	treatment	service	manpower	treatment	hospitalization	virus
6	program	program	program	treatment	education	social worker	society
7	health	support	treatment	health	certificate	infection	Corona Virus
8	hospitalization	service	visiting nursing	medical examination	support	living	area
9	rehabilitation	education	support	support	task	service	infectious disease
10	manpower	long-term	support	home-living	health	area	hospitalization
**Rank**	**April 2020**	**May 2020**	**June 2020**	**July 2020**	**August2020**	**September 2020**	
1	Corona	society	society	society	society	society	
2	hospitalization	Corona	dementia	welfare	welfare	welfare	
3	infection	treatment	welfare	service	support	dementia	
4	treatment	service	social worker	dementia	dementia	Corona	
5	society	hospitalization	service	social worker	service	treatment	
6	Corona19	dementia	long-term	certificate	social worker	education	
7	health	welfare	support	support	education	support	
8	dementia	support	treatment	treatment	treatment	service	
9	manpower	social worker	Corona	hospitalization	certificate	long-term	
10	area	long-term	health	education	home-living	social worker	

## Data Availability

Restrictions apply to the availability of these data. Data were obtained from Sometrend and are available at https://some.co.kr/ with the permission of Sometrend, (accessed on 16 April 2021).

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
