# Peer review of "Perceptions Related to Nursing and Nursing Staff in Long-Term Care Settings during the COVID-19 Pandemic Era: Using Social Networking Service"

_ijerph, 2021, doi:10.3390/ijerph18147398_

Round 1

Reviewer 1 Report

The authors have submitted an interesting study on the social perception of nurses' work in the Long-Term Setting during the COVID-19 Pandemic. However, the text cannot be published in its current version. The reviewer will comment on the points to improve following the order of the sections of the manuscript. Abstract section The authors write "Methods: This study used a cross-sectional design to investigate perceptions related to nursing and nursing staff in NHs during the COVID-19 pandemic era "The reviewer considers that expressed in this way, the meaning is unclear and recommends that the authors comment on the data source for the study in this same sentence. Introduction section Page 3 lines 97-98: Again the reviewer considers that the way of expressing the type of study is confusing and the reader does not obtain more information until he reads the research purpose. The reviewer suggests to the authors that "the perception of nursing staff in NHs ...." should be accompanied by the source of the data. Methods section Subpart Data Analysis The authors do not describe any statistical analysis but in the discussion section they speak of significant differences. Did the authors perform any statistical procedures?
Likewise, if the authors performed any statistical procedure, the p-value should appear in the corresponding tables and figures. Discussion section. the reviewer suggests authors not to start this section with a single sentence paragraph. Typically, the first paragraph summarizes the key points of the discussion. Pages 12, lines 297-299, the authors write "no significant change" twice but in the results there is no statistical technique performed that allows establishing statistical differences. Subsection limitations The reviewer considers that the authors could revise this subsection. The design has a risk of bias that the authors do not specify. The conclusions subsection is excessively long. The reviewer advises the authors to improve conciseness and clarify the relevant aspects that this study contributes.

Author Response

June, 21, 2021

RE: Revision of Manuscript #IJERPH-1268486

To whom it may concern.

Attached please find an electronic copy of a revision of #IJERPH-1268486, entitled "Perceptions Related to Nursing and Nursing Staff in Long-Term Care Settings During the COVID-19 Pandemic Era: Using Social Networking Service." In your editorial decision letter, you observed that the reviewers found considerable merit in the manuscript, but a number of issues required attention. You invited a minor revision and resubmission of the manuscript. We found your comments and those provided by the reviewers to be extremely helpful, and we hope you will find the revised manuscript to be substantially improved. Below, we provide a detailed point-by-point description of our response to each issue the reviewers raised (with comments made by the reviewers in italic font).

  1. This study addressed the issue regarding the important role of nursing and nursing staff in long term care setting by analyzing the messages posted in the social media channels 6 month before and after the declaration of COVID-19 outbreak in March, 2020. The authors concluded with two important points: 1. the public’s perception of importance of nursing staff in long term care setting enhanced after the outbreak of COVID-19 pandemic. 2. Need to improve the working environment and treatment of NH nurses. The strength is there’s no social analytics related to COVID-19 focusing on “the importance of nursing staff in long term care” yet. The weakness of this study is that it reflected the general perception of this issue and there’s lacking of in-depth analysis.

Yes, I agree with you. I added the risk of bias in this design in the limitation section and suggested that further research on the roles of nursing staff handling infection control in long-term care settings is needed.

  1. In overall, this is an observation and descriptive study with the exploration of large-scale of social media data. Therefore, the result and interpretation was straightforward. There’s no explicit research questions and hypotheses to be tested.

Yes, I agree with you. I revised the results and discussion sections.

  1. In table 4, the negative emotions exceed the positive emotions only appeared in March 2020 during the study period. After that, the positive emotions exceed the negative emotions again and lasted to the end of this study, how to interpret this phenomena? Is there any massive-infection and high mortality in nursing homes happened after March 2020?

Yes, I agree. The results can be confusing. Although massive-infection continuously happened after March 2020, the first major outbreak of massive-infection in nursing homes occurred in March 2020. Therefore, the number of documents, including the negative phrase “be infected,” sharply increased in March 2020. Since then, the number of such documents was not as high as it was in March; however, the contents of negative words before the pandemic were replaced with those related to infection. So, the proportion of negative words remained similar to before. I would like to say that the contents of negative words are important, not the proportion of negative words. I added this part on Page 12 lines 308-314.

  1. There’s no analysis to support for the authors’ second points of conclusion - Need to improve the working environment and treatment of NH nurses. How this conclusion was generated by the data? Please specify.

Yes, I totally agree. I clarified this point of conclusion.

  1. The large amount of elderly patients dwelling in the NHs got infected in USA, European country, and the high mortality of this population, how this study can incorporate the public’s opinions and link to the policy making for protection of this population and the nursing staff working in NHs.

Yes, I agree with you. I clarified this point of discussion. Through social big-data analysis, we identified that the public shared concerns about mass infection in nursing homes and with nursing staff in nursing homes regarding the COVID-19 crisis. To solve these problems, we can propose policies such as systemic infection-prevention education and legal staffing levels in nursing homes. I added this point on Page 13 lines 368-371.

  1. For the emotions analysis, some of the terms are closed to “facts” or “neutral” such as “infected”, “infection”, they are coded as negative emotions in this study. Please provide the guideline for classification of “positive emotions”, “negative emotions” or “neutral.

Yes, I totally agree. The classification was done through the sentiment dictionary, developed by Sometrend, which I used in this study for data analysis. Sometrend’s dictionary was developed based on frequently used contexts according to verbal-analytic techniques. I added this point on Page 4 lines 154-156.

I believe we have addressed each issue raised by the reviewers and hope you find the revisions to be satisfactory. We look forward to hearing from you. Thank you for your time and consideration.

Sincerely, Juh Hyun Shin, Associate Professor

Ewha Womans University, College of Nursing

Reviewer 2 Report

  1. This study addressed the issue regarding the important role of nursing and nursing staff in long term care setting by analyzing the messages posted in the social media channels 6 month before and after the declaration of COVID-19 outbreak in March, 2020. The authors concluded with two important points: 1. the public’s perception of importance of nursing staff in long term care setting enhanced after the outbreak of COVID-19 pandemic. 2. Need to improve the working environment and treatment of NH nurses. The strength is there’s no social analytics related to COVID-19 focusing on “the importance of nursing staff in long term care” yet. The weakness of this study is that it reflected the general perception of this issue and there’s lacking of in-depth analysis.
  2. In overall, this is an observation and descriptive study with the exploration of large-scale of social media data. Therefore, the result and interpretation was straightforward. There’s no explicit research questions and hypotheses to be tested.
  3. In table 4, the negative emotions exceed the positive emotions only appeared in March 2020 during the study period. After that, the positive emotions exceed the negative emotions again and lasted to the end of this study, how to interpret this phenomena? Is there any massive-infection and high mortality in nursing homes happened after March 2020?
  4. There’s no analysis to support for the authors’ second points of conclusion - Need to improve the working environment and treatment of NH nurses. How this conclusion was generated by the data? Please specify.
  5. The large amount of elderly patients dwelling in the NHs got infected in USA, European country, and the high mortality of this population, how this study can incorporate the public’s opinions and link to the policy making for protection of this population and the nursing staff working in NHs.
  6. For the emotions analysis, some of the terms are closed to “facts” or “neutral” such as “infected”, “infection”, they are coded as negative emotions in this study. Please provide the guideline for classification of “positive emotions”, “negative emotions” or “neutral”.

Author Response

June, 21, 2021

RE: Revision of Manuscript #IJERPH-1268486

To whom it may concern.

Attached please find an electronic copy of a revision of #IJERPH-1268486, entitled "Perceptions Related to Nursing and Nursing Staff in Long-Term Care Settings During the COVID-19 Pandemic Era: Using Social Networking Service." In your editorial decision letter, you observed that the reviewers found considerable merit in the manuscript, but a number of issues required attention. You invited a minor revision and resubmission of the manuscript. We found your comments and those provided by the reviewers to be extremely helpful, and we hope you will find the revised manuscript to be substantially improved. Below, we provide a detailed point-by-point description of our response to each issue the reviewers raised (with comments made by the reviewers in italic font).

Reviewer 2.

The authors have submitted an interesting study on the social perception of nurses' work in the Long-Term Setting during the COVID-19 Pandemic. However, the text cannot be published in its current version. The reviewer will comment on the points to improve following the order of the sections of the manuscript. Abstract section The authors write "Methods: This study used a cross-sectional design to investigate perceptions related to nursing and nursing staff in NHs during the COVID-19 pandemic era "The reviewer considers that expressed in this way, the meaning is unclear and recommends that the authors comment on the data source for the study in this same sentence.

->Yes, I totally agree with you. I added the concrete data sources in the abstract section.

 Introduction section Page 3 lines 97-98: Again the reviewer considers that the way of expressing the type of study is confusing and the reader does not obtain more information until he reads the research purpose. The reviewer suggests to the authors that "the perception of nursing staff in NHs ...." should be accompanied by the source of the data.

->Yes, I totally agree with you. I added the concrete data sources in the abstract section.

Methods section Subpart Data Analysis The authors do not describe any statistical analysis but in the discussion section they speak of significant differences. Did the authors perform any statistical procedures? Likewise, if the authors performed any statistical procedure, the p-value should appear in the corresponding tables and figures.

-> I agree with you. The phrase I used could be misleading. I simply compared the results before and after the pandemic, not using any statistical method. I revised it.

Discussion section. the reviewer suggests authors not to start this section with a single sentence paragraph. Typically, the first paragraph summarizes the key points of the discussion.

->I totally agree with you. I revised it.

 Pages 12, lines 297-299, the authors write "no significant change" twice but in the results there is no statistical technique performed that allows establishing statistical differences.

-> I agree with you. The phrase I used could be misleading. I simply compared the results before and after the pandemic, not using any statistical method. I revised it.

Subsection limitations The reviewer considers that the authors could revise this subsection. The design has a risk of bias that the authors do not specify.

->I totally agree with you. I added the risk of bias in this design in the limitation section.

 The conclusions subsection is excessively long. The reviewer advises the authors to improve conciseness and clarify the relevant aspects that this study contributes.

->I revised the conclusion section, focusing on the relevant aspects of this study.

I believe we have addressed each issue raised by the reviewers and hope you find the revisions to be satisfactory. We look forward to hearing from you. Thank you for your time and consideration.

Sincerely, Juh Hyun Shin, Associate Professor

Ewha Womans University, College of Nursing

Round 2

Reviewer 1 Report

the authors have made an important effort to improve the quality of the manuscript.

Reviewer 2 Report

The authors have improved this manuscript. It can be published in the present form.